# SARS-CoV-2 Seroprevalence among Healthcare Workers after the First and Second Pandemic Waves

**DOI:** 10.3390/v14071535

**Published:** 2022-07-14

**Authors:** Nathalie de Visscher, Xavier Holemans, Aline Gillain, Anne Kornreich, Raphael Lagasse, Philippe Piette, Manfredi Ventura, Frédéric Thys

**Affiliations:** 1Department of Infectiology and Internal Medicine, Grand Hôpital de Charleroi (GHdC), B-6000 Charleroi, Belgium; nathalie.devisscher@ghdc.be (N.d.V.); xavier.holemans@ghdc.be (X.H.); 2Clinical Research and Translational Unit, Grand Hôpital de Charleroi (GHdC), B-6000 Charleroi, Belgium; aline.gillain@ghdc.be; 3Department of Laboratory Medicine, Grand Hôpital de Charleroi (GHdC), B-6000 Charleroi, Belgium; anne.kornreich@ghdc.be; 4Department of Medico-Economic Information, Grand Hôpital de Charleroi (GHdC), B-6000 Charleroi, Belgium; raphael.lagasse@ghdc.be (R.L.); philippe.piette@ghdc.be (P.P.); 5Department of Information Technologies, Grand Hôpital de Charleroi (GHdC), B-6000 Charleroi, Belgium; 6School of Public Health, Université Libre de Bruxelles (U.L.B.), B-1070 Brussels, Belgium; 7Department of Medical Management Team, Grand Hôpital de Charleroi (GHdC), B-6000 Charleroi, Belgium; manfredi.ventura@ghdc.be; 8Department of Acute and Emergency Medicine, Grand Hôpital de Charleroi (GHdC), B-6000 Charleroi, Belgium; 9Continuing Education Unit UCLouvain Woluwe, Université Catholique de Louvain, B-1200 Woluwe-Saint-Lambert, Belgium and CEM-ETHICS EA 7446, Université Catholique de Lille, F-59000 Lille, France

**Keywords:** healthcare workers, SARS-CoV-2, seroprevalence, pandemic waves, Belgium

## Abstract

**Background:** The Grand Hôpital de Charleroi is a large non-academic Belgian hospital that treated a large number of COVID-19 inpatients. In the context of this pandemic, all professions-combined healthcare workers (HCWs), and not only direct caregivers, are a frontline workforce in contact with suspected and confirmed COVID-19 cases and seem to be a high-risk group for exposure. The aim of our study was to estimate the prevalence of anti-SARS-CoV-2 antibodies in HCWs in our hospital after the first and second pandemic waves and to characterize the distribution of this seroprevalence in relation to various criteria. **Methods:** At the end of the two recruitment periods, a total of 4008 serological tests were performed in this single-center cross-sectional study. After completing a questionnaire including demographic and personal data, possible previous COVID-19 diagnostic test results and/or the presence of symptoms potentially related to COVID-19, the study participants underwent blood sampling and serological testing using DiaSorin’s LIAISON^®^ SARS-CoV-2 S1/S2 IgG test for the first phase and LIAISON^®^ SARS-CoV-2 TrimericS IgG test for the second phase of this study. **Results:** In total, 302 study participants (10.72%) in the first round of the study and 404 (33.92%) in the second round were positive for SARS-CoV-2-IgG antibodies. The prevalence of seropositivity observed after the second wave was 3.16 times higher than after the first wave. We confirmed that direct, prolonged, and repeated contact with patients or their environment was a predominant seroconversion factor, but more unexpectedly, that this was the case for all HCWs and not only caregivers. Finally, the notion of high-risk contact seemed more readily identifiable in one’s workplace than in one’s private life. **Conclusions:** Our study confirmed that HCWs are at a significantly higher risk of contracting COVID-19 than the general population, and suggests that repeated contacts with at-risk patients, regardless of the HCWs’ professions, represents the most important risk factor for seroconversion (Clinicaltrials.gov number, NCT04723290).

## 1. Introduction

Coronaviruses have been circulating in the population for a long time and are associated with rather mild respiratory symptoms. The new SARS-CoV-2 belongs to the beta-coronavirus subfamily and is closely genomically related to SARS-CoV-1 that caused the SARS epidemic in 2002–2003. Severe acute respiratory syndrome coronavirus-2 (SARS-CoV-2) and coronavirus disease 2019 (COVID-19) emerged in December 2019 from Wuhan, Hubei Province, China, and the World Health Organization (WHO) declared it a pandemic situation on 11 March 2020 [1]. As of 19 November 2021, WHO reported globally 254,847,065 confirmed cases and 5,120,712 total deaths due to COVID-19 [2]. On the same date, the Belgian Institute for Public Health (Sciensano) reported 1,559,998 confirmed cases, 86,444 hospital admissions, and 26,526 deaths in Belgium [3].

In this unprecedented pandemic context, all professions’ combined healthcare workers (HCWs), and not only direct caregivers, are a frontline workforce in contact with suspected and confirmed COVID-19 cases and seem to be a high-risk group for exposure. Several recent meta-analyses have shown that the prevalence of anti-SRAS-CoV-2 antibody positive HCWs was high [4,5]. In our setting, though several HCWs showed symptoms suggestive of COVID-19 at the beginning of the epidemic, they were unable to get tested because PCR testing was limited to hospitalized patients in Belgium due to the scarcity of supplies. Furthermore, even when the PCR tests are available, they have a limited sensitivity [6]. Indeed, pre-analytical factors such as suboptimal sampling, variation in viral shedding and variable time span between the onset of symptoms and time of sampling can impact the sensitivity of the PCR tests [7]. Additionally, the interpretation of test results should take into account pretest probability of infection for asymptomatic and symptomatic individuals [8]. That is why people with a high clinical suspicion of COVID-19 should ideally be retested the same day or on subsequent days if the initial results are negative by a molecular SARS-CoV-2 assay to limit false negative results [9].

However, in our setting, at the early stages of the pandemic, this was not feasible, and, therefore, we relied upon the detection of anti-SARS-CoV-2 antibodies in individuals who previously presented suggestive symptoms related to SARS-CoV-2 infection to confirm the presumptive diagnosis of COVID-19. Anti-SARS-CoV-2 antibody tests are known to be accurate in detecting previous SARS-CoV-2 infection if performed more than 14 days after the onset of symptoms but are much less sensitive if performed earlier [10,11]. Consequently, we assume that studying the evolution of the prevalence of anti-SARS-CoV-2 antibodies among HCWs is important to understand the spread of COVID-19 in hospitals and to evaluate the success of preventive and protective measures implemented in hospitals, which could afterwards be extended to the general population. Furthermore, in terms of crisis management, it is also useful to know the proportion of workers who have already been in contact with the virus in order to optimize the organization of care teams for the benefit of all hospitalized patients, whether or not they are infected with the virus.

This study seeks to evaluate the anti-SARS-CoV-2 seroprevalence of HCWs of the Grand Hôpital de Charleroi (GHdC) at two key moments of the COVID-19 pandemic: after the first wave and after the second wave. The GHdC is a large non-academic institution that was the second hospital in Belgium, and the first in the Walloon Region, to admit the largest number of COVID-19 patients. In addition to studying our HCWs’ anti-SARS-CoV-2 seroprevalence, we will also analyze the distribution of this large group of hospital professionals according to age, gender, occupation, and geographical location (our care institution being spread over several sites). We will match the HCWs’ serological results with their previous COVID-19 history, symptoms, and possible known contacts with COVID-19 infected individuals in an effort to contribute to a better description and understanding of the impact of this pandemic within the hospital. We will also analyze the male/female ratio of the participants to address the pending question of possible gender-specific immune responses [12].

## 2. Materials and Methods

### Study Design

Our study was conducted during two distinct inclusion periods: one after the first wave of the pandemic (summer 2020) and the other after the second wave (end of winter 2021). In both recruitment periods, the safety procedures following the Belgian guidelines (Sciensano.be) were applied according to the availability of protective equipment made available to all staff through the constant efforts and foresight of the hospital and all HCWs at the Grand Hôpital de Charleroi were invited to participate in this monocentric cross-sectional study. The two vaccination campaigns were organized by the Belgian Health Authorities. They decided to first vaccinate the healthcare workers in hospitals and then staff in retirement homes. Belgian early vaccination started in January 2021, GHdC in February 2021, following guidelines and access to vaccines and deliveries.

At GHdC, the vaccination was administrated first to healthcare workers directly in contact with COVID-19-positive patients. This was gradually extended to the other members of the institution. Vaccine administration and the participation in serological testing were voluntary and independent for each GHdC healthcare worker.

None of the HCWs included had received vaccination against COVID-19 prior to their blood being drawn for serological testing. After having given their consent, all participants were asked to fill out a questionnaire which included demographic and personal data, possible prior results of a COVID-19 PCR diagnostic test and/or the presence of symptoms suggestive of a SARS-CoV-2 infection, and knowledge of prior contact with COVID-19-infected individuals, whether at work or outside of work. No other questions about prior medical history were asked. Serological sampling and testing were carried out after collecting the completed questionnaires using the DiaSorin LIAISON^®^ SARS-CoV-2 S1/S2 IgG test for the first phase and the DiaSorin LIAISON^®^ SARS-CoV-2 TrimericS IgG test for the second phase of this study. At the time of the transition between the two kits, a concordance study was performed on 521 samples from the second phase of the study (259 negative and 262 positive sera, as determined with the DiaSorin LIAISON^®^ SARS-CoV-2 S1/S2 IgG test) and showed no difference in the interpretation of serological results (unpublished results). Indeed, the concordance between the two kits was measured at 94% and two “false negative” and 14 “false positive” results were detected with the new DiaSorin LIAISON^®^ SARS-CoV-2 TrimericS IgG test, bringing the overall positivity rate in this subsample from 50.3% to 48.0% (*p* < 0.01). Thus, even though DiaSorin’s new kit significantly decreases the positivity rate by a factor of 0.954, it does not significantly affect the comparison of the seroconversion rates of our study population between the two phases. Thus, even though DiaSorin’s new kit purportedly offers an improved detection of IgG neutralizing antibodies, it does not significantly alter the seroconversion rate of our study population in the second phase.

Consent forms, participant information, questionnaires, and test requests were centralized via a highly secure IT platform to ensure data confidentiality throughout the process.

The procedure for healthcare workers who tested positive for SARS-CoV-2 between 20 June and 11 August 2020 and during 6 February to 31 March 2021 followed the Belgian guidelines (Sciensano.be). During the first period, they were isolated for at least 7 days at home and for at least 10 days during the second period due to the Delta COVID-19 variant. Return to work was allowed on the condition that there was no fever for at least 3 days and that there was an improvement in the respiratory symptoms.

The descriptive statistical evaluation of the questionnaire data, correlated with the laboratory results, used the following methods as appropriate: computation of the 95% confidence limits for proportions using the Wilson method [13] and uncorrected Pearson’s Chi2 test for comparing proportions in independent samples (with two-tailed probability) [14]. The study protocol was fully approved by the hospital’s independent ethics committee and the Clinicaltrials.gov Identifier is NCT04723290.

## 3. Results

For the first phase of the study, from 20 June to 11 August 2020, 3474 HCWs expressed their interest and filled out the informed consent form but finally, only 2817 of them came to be serologically tested. During the second phase of the study, from 6 February to 31 March 2021, 1191 HCWs were serologically tested. Thus, over both rounds of this study, we collected 4008 serology results for further analysis. Table 1 presents the distribution of the participants by gender, age, and profession category within the hospital for each round. During the first phase of this study, the gender distribution of our sample was 2293 females (81.4%) and 524 males (18.6%) and perfectly matched that of the general hospital staff, i.e., 3201 females (82%) and 693 males (18%). That same concordance was observed for the age distribution and that was also the trend for the second phase of this study.

Although all professions were represented in the participating HCWs, caregivers, mainly nurses and doctors, were the largest group (63.1% after the first pandemic wave versus 53.7% after the second wave). No significant difference was observed for the caregiving function according to gender. Anti-SARS-CoV-2-IgG antibodies were detected in 302 (10.72%) of the 2817 subjects included during the first phase of the study and in 404 (33.92%) of the 1191 subjects included during the second phase; thus, showing a 3.16-fold increase in seropositivity of our HCWs (see Table 2). If during the first round of the study seropositivity was more prevalent in women with a ratio of 1.21, this trend was reversed in the second round, with a slightly higher, though not significant, seropositivity observed in men with a ratio of 1.13. In the first round, difference in age-related seropositivity was statistically significant (*p* < 0.01), ranging from 5% in the 60–69 age group to 15% in the younger age group (20–29 years), whereas in the second round, no statistically significant difference (*p* = 0.06) was observed between the lowest seropositivity prevalence observed in the 40–49 age group and the highest one in the younger 20–29 age group. In both rounds, our sample of participants above the age of 69 was too limited to draw any conclusions. The hypothesis that HCWs are a population at high risk of contracting SARS-CoV-2 infection as a result of patient contact is supported by the following observations from our study: hospital professions who have structurally more numerous, closer, and prolonged contacts with patients or their fluid samples, such as caregiver functions, demonstrated a statistically significant difference in terms of seroconversion (RR 2.3 in the first round (*p* < 0.001) and RR 1.84 in the second round (*p* < 0.0001)). Unexpectedly, after the first wave of the pandemic, we observed that the hospital’s technical staff, who had been heavily involved in the many processing jobs in the COVID-19 units, also had a significant seroconversion rate. However, this observation was not reiterated after the second wave. To our current knowledge of the literature, this significant seroconversion rate of ancillary staff of COVID-19 units is original and reported for the first time in this pandemic context.

In both rounds of our study, participants were asked if they had had close contact with COVID-19-positive people at work or in their private lives, with no time limit for the first phase and going back three months for the second phase (Table 3). In the first phase of our study, most participants could not easily provide this information, probably due to the scarcity of PCR tests at that time, whereas in the second phase of the study, more participants could identify this type of contact. In the first round, only 188 study participants (6.7%) reported close contact with a case of COVID-19 outside of work, and among these, 17% tested positive for anti-SARS-CoV-2 antibodies, whereas the rest of the study population, who answered “no” or “unknown” to this question, showed a 10.3% seropositivity rate (*p* = 0.004). In the second round, these reports increased to 270 study participants (22.67%) among whom 45% were seropositive. More surprisingly, even though close contacts with COVID-19 patients in the professional setting were more readily identifiable during the first round, with 1054 respondents (37.4%) reporting them, the seropositivity rate (17.6%) was similar in this group to the one in the private setting. The findings were similar in the second round where a high-risk contact was reported by 270 respondents (22.7%) outside of work and by 288 respondents (24.2%) at work, and both groups had a similar seroconversion rate of 45.2% and 47.2%, respectively.

Finally, in both rounds of the study, participants were asked if they had previously presented symptoms suggestive of COVID-19 (Table 4). In the first round, 161 participants (5.7%) answered “yes”, though only 133 of them were confirmed with a PCR test. Those HCWs showed 79.5% seropositivity, whereas the 1264 who declared not having had symptoms of COVID-19 had a seropositivity rate of 4%, which may correspond to asymptomatic forms of the disease. In the second round, it was interesting to observe that the 290 (24.4%) HCWs who reported having had prior symptoms suggestive of COVID-19 had a significantly higher seropositivity rate than those who had had no symptoms (87.6%; *p* < 0.0001).

## 4. Discussion

The main aim of this study was to determine the anti-SARS-CoV-2 seroprevalence in healthcare workers at the Grand Hôpital de Charleroi, Belgium, at two key moments in time, i.e., after the first and the second pandemic waves. Its secondary objective was to examine the distribution of this anti-SARS-CoV-2 prevalence by gender, age, and type of profession, but also in relation to possible previous symptoms suggestive of COVID-19 and the notion of high-risk contact with an infected person in a large group of healthcare workers. Our study is one of the few published studies on the seroprevalence of SARS-CoV-2 in all professions employed by a hospital and shows a relatively high proportion of HCWs who are IgG positive for SARS-CoV-2 compared to other centers in Europe.

Overall participation was high (75% of all hospital staff) and fully representative of our staff in terms of gender and age distribution in both rounds of the study. All hospital professions were represented in our overall study sample, although the proportion of direct caregivers (mainly doctors and nurses) was higher than the rest.

No important bias was noticed in the sampling for the first phase of the study, and we may consider that the results presented adequately reflect the global situation in the hospital. For the second phase, a significant deficit in the participation of caregivers probably led to an underestimation of the seropositivity prevalence in our sample. In the absence of this bias, we assume that the global difference between the seropositivity rates of both phases would have been even greater than what was observed.

A lack of reliability in the answers of the participants concerning exposure to contacts and history of symptoms suggestive of COVID-19 cannot be excluded, and probably leads to an underestimation of both these individual characteristics. However, the confounding impact of this recall bias on the analytical results of seropositivity can be ruled out due to the information’s collection chronology and its random distribution in the comparison groups. Seropositivity rate for SARS-CoV-2-IgG antibodies was 10.72% (302 among 2817 study participants) after the first pandemic wave, and 33.92% (404 among 1191 study participants) after the second pandemic wave. The seroprevalence in our cohort thus showed a 3.16-fold increase in the second round of our study. We confirmed that repeated, direct, and prolonged contact with patients or their environment was a significant seroconversion factor. Finally, the notion of high-risk contact seemed easier to identify in the professional setting than in the private setting and the identification of high-risk contact was easier during the second wave in both settings.

The COVID-19 pandemic is arguably the most important and rapidly evolving infectious public health issue in the world since the AIDS pandemic, and worldwide efforts are made to suppress or eliminate its spread by all means including hand hygiene, social distancing, early screening, diagnosis, and quarantine [15,16]. In this context, knowledge of the SARS-CoV-2 seroprevalence of the population, combined with data from COVID-19 diagnostic tests, are useful tools to assess the proportion of the population infected with SARS-CoV-2 in a recent or somewhat delayed manner. These results should make it possible to calibrate responses to the waves of this pandemic and guide policy makers in the fight against it. A recent meta-analysis of SARS-CoV-2 seroprevalence in the world population has shown that although it is geographically heterogeneous, ranging from 0.37% to 22.1%, the absence of active immunization is estimated at around 3.38% overall [17]. Another recent study in Croatia analyzed the seroprevalence and the neutralizing antibody response in the general Croatian population after the first and second pandemic waves. This study showed a significant difference in the overall seroprevalence rate after the first wave (ELISA 2.2%, Virus Neutralization Test (VNT) 0.2%) and second wave (ELISA 25.1%, VNT 18.7%) [18]. The knowledge of the seroprevalence in healthcare workers is also important, both for the organization of care within the hospital and for the implementation of global public health policies. The prevalence of SARS-CoV-2 antibodies among HCWs is shown to be high. After the first wave, a large systematic review and meta-analysis of 127,480 healthcare workers estimated an overall seroprevalence of 8.7% [5]. This figure was slightly lower than previous meta-analyses, which estimated it at 10.1% [4,19]. Seroprevalence varied by region, with 12.7% in North America, 8.5% in Europe, 8.2% in Africa, and 4% in Asia [5]. After the second wave, several authors reported higher overall prevalence rates among HCWs in different parts of the world: 20.8% in Kenya [20], 23.65% in Ahmedabad (India) [21], and even 31.6% in Spain [22]. In our study, the measured seroprevalence was 10.72% after the first wave, which was higher than the values reported in the various studies previously carried out after the first wave in Belgium (that ranged between 6.4% and 7.7% [23,24]), and 33.92% after the second wave. This difference could be due to the fact that the province of Hainaut, where our hospital is located, is one of the Belgian provinces that bore the heaviest burden of COVID-19 infections in Belgium. This explanation is supported by the findings that seroprevalence rates in HCWs reported in the literature are proportional to levels of exposure to SARS-CoV-2 [25,26]. On this matter, Prakash O. et al. demonstrated that the seropositivity among healthcare workers by area in India was closely correlated with the number of cases reported in the respective area. Surprisingly, they also found that the rate of seropositivity among healthcare workers in the oldest and most severely affected areas was lower than in more recently affected areas. The authors speculated that IgG antibodies may not be lasting [16]. Another significant factor to consider is the timing of the test in relation to the course of the epidemic wave. It is noteworthy that some recent publications observed variations in the seroprevalence of anti-SARS-CoV-2 antibodies among different groups of healthcare workers. The highest seroprevalence was observed among those working in acute or emergency medicine or in general internal medicine, while the lowest seroprevalence was observed among those working in intensive care medicine [27]. These findings support the hypothesis that the different risks of exposure to SARS-CoV-2 exist in the hospital environment. Intensive care units have often been pointed as high-risk environments and have therefore benefited from the use of optimized personal protective equipment (PPE). Furthermore, in a COVID-19 unit, the caregivers wear the same PPE as they go from one patient to the next. That is not the case in an emergency department, where the caregivers constantly change their PPE between patients so as to limit the risk of spreading the disease. In our institution, we observed the same phenomenon with the dedicated COVID-19 intensive care or hospitalization units. Surprisingly, a subgroup from the first round, the technical staff of the hospital, showed a higher proportion of seroconversion than that of certain subgroups of caregivers (see Table 2). This group was heavily involved in the transformation work needed to set up the COVID-19 units. We assumed that they were subjected to an environment with high exposure to the virus and that the specificity of their work required them to frequently change their PPE when they had to take over new tools or materials for their technical work. These observations tend to show that frequently changing PPE in a potentially contaminated environment is in itself a contamination risk, and even more so if professional training for this is insufficient or non-existent. The fact that we did not observe this increased seroconversion rate in this subgroup in the second round of our study reinforces our hypothesis, since most of the transformation work took place during the first wave and that, by the second wave, staff training for changing PPE was improved. This combination of factors could explain this original observation.

It has been reported that gender may impact immune responses to COVID-19 [28]. Gender-related differences at several levels (infection and severity) are reported in the literature [29,30]. We therefore also wanted to address this gender issue in our study (see Table 2). Although we found a higher seroprevalence among women in the first round of our study, we were not able to demonstrate a significant difference for this parameter, but we must keep in mind that our univariate approach did not consider other exposure factors. The reversal of this finding in the second round of our study, though still not statistically significant, with a higher seroprevalence among men, led us to reject the impact of gender on seroprevalence for our cohort [31].

Lastly, we showed that, in the first round of our study, a high-risk contact was easier to confirm in the workplace than in the private setting, though this was still difficult for healthcare workers due to the scarcity of PCR tests, especially for outpatients. The increasing availability over time of PCR tests and the implementation of the contact-tracing scheme allowed a better assessment of risk-related contacts thereafter [32]. Despite those difficulties, it is noteworthy that among HCWs who declared having had a high-risk contact, the seroconversion rate was similar in both non-work and work settings (17.02% vs. 17.55% in the first round and 45.19% vs. 47.22% in the second round). This leads us to believe that high-risk contacts were underestimated and/or the transmission rate was low during the first wave, compared with the second wave, where high-risk contacts were better identified and the transmission rate was increased with the emergence of the alpha-variant of the virus. Interestingly, HCWs who declared not having had a high-risk contact or not knowing, also had a similar seroconversion rate in both non-work and work settings, especially in the second round of this study (30.62% vs. 29.68%); thus, indicating that asymptomatic transmission was substantial in both settings and underlining the importance of applying barrier measures at all times. It should be noted that a meta-analysis estimated that nosocomial transmission was the source of SARS-CoV-2 infection in approximately 44% of cases [33]. For this reason, some authors suggested regular anti-SARS-CoV-2 IgG screening of healthcare workers to better identify risk factors and adapt the organization of care. It was demonstrated that healthcare workers who care for COVID-19 patients and identify a high-risk contact have a higher seroprevalence [20]. The identification of such a contact should lead to the systematic use of adequate PPE in order to limit the risk of contamination.

In conclusion, though this study has the limitation of being monocentric, it has the advantage of pinpointing the time between possible exposure to SARS-CoV-2 and seroconversion across a wide variety of professions within our hospital, giving a more accurate estimation of seroprevalence among our HCWs, which is indeed slightly higher than what has been reported in previous studies conducted in our country and in Europe during both pandemic waves. It confirmed that healthcare workers represent a population that is at significantly higher risk of contracting COVID-19 than the general population. We suggest that the notion of repeated contact with high-risk patients and the regular change in PPE represent the most important risk factors for seroconversion. In this context, absolute adherence to infection prevention and barrier measures, sufficient and adequate personal protective equipment, as well as early recognition, identification, and isolation of healthcare workers infected with SARS-CoV-2 remain mandatory to reduce the risk of COVID-19 infection and transmission, especially since the notion of high-risk contact, although more readily accessible in the professional than in the private setting, continues to be elusive.

## Figures and Tables

**Table 1 viruses-14-01535-t001:** Demographic characteristics, contact with patients, and profession among healthcare workers involved in the survey: first and second rounds.

	Summer 2020		Winter 2021 (February–March)
	Participants of the Survey (N = 3474)	Participants with Serologic Test (N = 2817)	Participants with Serologic Test before Immunization (N = 1191)
	%	%	%
**Gender**			
Females	81.4	81.4	82.7
Males	18.6	18.6	17.3
**Age Category**			
<20	0.1	0.1	0.1
20–29	16.4	15.5	13.1
30–39	24.2	24.8	21.8
40–49	23.7	24.1	24.9
50–59	27.7	27.9	30.1
60–69	7.8	7.6	9.7
70–79	0.2	0.1	0.3
**Contact with patients**			
No	42.9	42.6	54.2
Yes	57.1	57.4	45.9
**Profession**			
Nurse and Nurse Aide	37.3	37.5	25.7
Physician	14.8	14.8	12.3
Other Caregiver	10.5	10.8	15.7
**Caregivers sub-total**	62.6	63.1	53.7
Administrative	15.2	16.1	25.6
Technical and Logistics	5.5	5.2	4.6
Other	16.7	15.7	16.1

**Table 2 viruses-14-01535-t002:** SARS-CoV-2 seropositivity among participating healthcare workers according to gender, age, contact with patients, and profession: situation corresponding to the two rounds of the survey.

SUMMER 2020							WINTER 2021							Ratio 2/1
		Seropositivity SARS-CoV-2							Seropositivity SARS-CoV-2					
	N	%	95% C.I.	Ratio	*p*			N	%	95 % C.I.	Ratio	*p*		
Total	2817	10.72	(9.63–11.92)	-	-		Total	1191	33.92	(31.29–36.66)	-	-		3.16
**Gender**							**Gender**							
Females	2293	11.08	(9.86–12.43)	1.21	NS		Females	985	33.20	(30.33–36.20)	1.00	NS		3.00
Males	524	9.16	(6.98–11.94)	1.00			Males	206	37.38	(30.75–44.37)	1.13			4.08
**Age Category**							**Age Category**							
<20	2	0	-	-	0.005		<20	1	0	-	-	NS		
20–29	436	14.91	(11.87–18.56)	1.58			20–29	156	44.23	(36.29–52.39)	1.46			2.97
30–39	698	9.89	(7.89–12.32)	1.05			30–39	260	33.08	(27.39–39.16)	1.09			3.34
40–49	678	9.44	(7.46–11.87)	1.00			40–49	297	30.30	(25.13–35.88)	1.00			3.21
50–59	786	11.7	(9.64–14.14)	1.24			50–59	358	35.20	(30.43–40.28)	1.16			3.01
60–69	213	5.16	(2.61–9.05)	0.55			60–69	116	27.59	(19.70–36.66)	0.91			5.35
70–79	4	25	(0.63–80.59)	-			70–79	3	33.33	(0.84–90.57)	-			
**Contact with patients**							**Contact with patients**							
No	1183	6.17	(4.94–7.69)	1.00	*p* < 0.0001		No	561	24.42	(21.05–28.14)	1.00	*p* < 0.0001		3.96
Yes	1594	14.3	(12.67–16.11)	2.32			Yes	475	45.05	(40.64–49.55)	1.84			3.15
**Trade**							**Trade**							
Nurse & Nurse Aide	1057	16.08	(13.99–18.42)	4.05			Nurse & Nurse Aide	306	48.04	(42.50–53.63)	2.19	*p* < 0.0001		2.99
Physician	418	9.81	(7.31–13.04)	2.47			Physician	146	43.15	(34.99–51.60)	1.96			4.40
Other Caregiver	303	9.24	(6.47–13.03)	2.33			Other Caregiver	187	24.60	(18.61–31.41)	1.12			2.66
Administrative	453	3.97	(2.53–6.19)	1.00	*p* < 0.0001		Administrative	305	21.97	(17.69–26.95)	1.00			5.53
Technical & Logistics	145	13.1	(8.08–19.7)	3.30			Technical & Logistics	55	34.55	(22.24–48.58)	1.57			2.64
Other	441	5.9	(4.05–8.5)	1.49			Other	192	32.29	(25.74–39.40)	1.47			5.47
**Caregivers sub-total**	1778	13.44	(11.94–15.11)	3.39	*p* < 0.0001		**Caregivers sub-total**	639	40.06	(36.33–43.91)		*p* < 0.0001		2.98

**Table 3 viruses-14-01535-t003:** SARS-CoV-2 seropositivity among participating healthcare workers according to close contact in the surrounding or at work: situation corresponding to the two rounds of the survey.

SUMMER 2020							WINTER 2021					
		Seropositivity SARS-CoV-2							Seropositivity SARS-CoV-2			
	N	%	95% C.I.	Ratio	*p*			N	%	95% C.I.	Ratio	*p*
**Close contact in the surrounding**							**Close contact in the surrounding**					
Yes	188	17.02	(11.94–23.17)	1.66	0.004		Yes	270	45.19	(48.67–60.85)	1.48	*p* < 0.0001
Unknown	2629	10.27	(9.17–11.49)	1.00			Unknown	921	30.62	(27.73–33.67)	1.00	
**Close contact at work**							**Close contact at work**					
No	1054	17.55	(15.37–19.97)	2.64	*p* < 0.0001		No	288	47.22	(41.34–53.16)	1.59	*p* < 0.0001
Unknown	1763	6.64	(5.57–7.9)	1.00			Unknown	903	29.68	(26.79–32.74)	1.00	

**Table 4 viruses-14-01535-t004:** Seropositivity according to the “symptoms suggestive of COVID-19” response.

Summer 2020						Winter 2021					
		Seropositivity						Seropositivity			
		SARS-CoV-2						SARS-CoV-2			
	**N**	%	95% C.I.	Ratio	*p*		**N**	%	95% C.I.	Ratio	*p*
**Symptoms suggestive of COVID-** **19**						**Symptoms suggestiveof COVID-19**					
Yes	161	79.5	(72.44–85.45)	20.08	*p* < 0.0001	Yes	290	87.59	(83.23–91.15)	6.77	*p* < 0.0001
No	1264	3.96	(3.01–5.18)	1		No	657	12.94	(10.59–15.72)	1	
Unknown	1392	8.91	(7.52–10.52)	2.25		Unknown	244	26.64	(21.20–32.65)	2.06	

## Data Availability

The data related to this study are available with access on request on the internal server of the clinical research unit and laboratory of the Grand Hôpital de Charleroi.

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
