# Peer review of "SARS-CoV-2 Seroprevalence among Healthcare Workers after the First and Second Pandemic Waves"

_viruses, 2022, doi:10.3390/v14071535_

Round 1

Reviewer 1 Report

The article presented for review is interesting and fits perfectly with the content of the special issue. The research was carried out on a large group of employees of a very large hospital. The article is methodologically correct and written in clear language. I have some minor comments: 1. Study design. What was the procedure for those who tested positive (were they sent to home isolation?)

2. Were the safety procedures the same in both analyzed periods?

3. I do not see in the text analyzes using the described statistical methods "The descriptive statistical evaluation of the questionnaire data, correlated with the laboratory results, used the following methods as appropriate: Chi2- Mann-Whitney - Wilcoxon Two-Sample Test and Kruskal-Wallis test." line 107 - 109.

4. What soft-ware were used forstatistical calculation? I checked few result with online calculator and got  slightly different values of IC.

  https://www.statology.org/confidence-interval-proportion-calculator/

Section: Results. in Table 2, the ratio 2/1 was calculated. Maybe the same ratio can be calculated for the results in tables 3 and 4.

Author Response

A1. Study design. What was the procedure for those who tested positive (were they sent to home isolation?)

“The procedure for healthcare workers who tested positive for SARS-CoV-2 between June 20 to August 11, 2020 and during February 6 to March 31, 2021 followed the Belgian guidelines (Sciensano.be). During the first period, they were isolated for at least 7 days at home and for at least 10 days during the second period due to the Delta COVID-19 variant.  Return to work was allowed on the condition that there was no fever for at least 3 days and that there was an improvement of the respiratory symptoms.”

We have incorporated this additional text from line 135 to line 140 in the chapter study design

A2. Were the safety procedures the same in both analyzed periods?

“The safety procedures followed the Belgian guidelines (Sciensano.be) were applied according to the availability of protective equipment made available to all staff through the constant efforts and foresight of the hospital. “

We have incorporated this additional text from line 98 to line 100 in the chapter study design

A3. I do not see in the text analyzes using the described statistical methods "The descriptive statistical evaluation of the questionnaire data, correlated with the laboratory results, used the following methods as appropriate: Chi2- Mann-Whitney - Wilcoxon Two-Sample Test and Kruskal-Wallis test." line 107 - 109.

We agree and have reworded the description by adding two bibliographic references for better understanding from line 132 to line 137 in the chapter study design

A4. What soft-ware were used for statistical calculation? I checked few result with online calculator and got  slightly different values of IC.

Thank you for this comment. You will find below all the explanations you need to answer your question.

Different methods have been proposed by statisticians to determine the confidence limits :

  • The exact methods, including the Fisher method
  • The Wald method
  • The Wilson method

The software used for statistical calculations is EpiInfoR , a statistical package widely used in the world and implemented first by WHO, and later by CDC Atlanta.

For computing the confidence limits for proportions, this package uses the Wilson method
(WILSON E. B., « Probable inference, the law of succession, and statistical inference », Journal of the American Statistical Association, n°22, 1927, pages 209-212.)
As a difference, the method used by the reviewer (  https://www.statology.org/confidence-interval-proportion-calculator/ ) uses the (more basic) Wald method (without correction).
(see for ex. VOLLSET S. E., « Confidence intervals for a binomial proportion », Statistics in Medicine, n°12(9), 1993, pages 809-824.)

The Wilson method appears to be more conservative than the Wald method (larger C.I.)
Many authors recommend to use the Wilson method, which is more performing than the classical one, and approximates efficiently the exact methods.

As an example : in Table 2, first line   N = 2817 and 10,72 % seropositivity     (à n=302) :

(treated here by another online available calculator : https://www.openepi.com/Proportion/Proportion.htm )

95% confidence limits for the proportion 302/2817

Multiply = 100

Large population size or sample with replacement.

CL low

per 100

CL high

Proportion in percent

10.7206

Mid-P Precise

9.618

11.9

Fisher exact method (Clopper-Pearson)

9.601

11.92

Wald test (approximate normal)

9.578

11.86

Modified Wald test (Agresti-Coull)

9.63

11.92

Result (Wilson)*

9.631

11.92

Continuity results

Correction (Fleiss quadratic)

9.614

11.94

A5 .Section: Results. in Table 2, the ratio 2/1 was calculated. Maybe the same ratio can be calculated for the results in tables 3 and 4.

We agree and have replaced table 3 with a new table (see below) which includes the calculation of the same ratio.  See line 210.

SUMMER 2020

WINTER 2021

Seropositivity
SARS-CoV-2

Seropositivity
SARS-CoV-2

N

%

95 % C.I.

Ratio

P

N

%

95 % C.I.

Ratio

P

Close contact in the surrounding

Close contact in the surrounding

Ratio 2/1

Yes

188

17.02

(11.94-23.17)

1.66

0.004

Yes

270

45.19

(48.67-60.85)

1.48

P<0.0001

2.66

Unknown

2629

10.27

(9.17-11.49)

1.00

Unknown

921

30.62

(27.73-33.67)

1.00

2.98

Close contact at work

Close contact at work

Yes

1054

17.55

(15.37-19.97)

2.64

P<0.0001

Yes

288

47.22

(41.34-53.16)

1.59

P<0.0001

2.69

Unknown

1763

6.64

(5.57-7.9)

1.00

Unknown

903

29.68

(26.79-32.74)

1.00

4.47

SUMMER 2020

WINTER 2021

Seropositivity

Seropositivity

SARS-CoV-2

SARS-CoV-2

N

%

95 % C.I.

Ratio

P

N

%

95 % C.I.

Ratio

P

Symptoms suggestive
of Covd-19

Symptoms suggestive
of Covd-19

Ratio 2/1 

Yes

161

79.5

(72.44-85.45)

20.08

0.004

Yes

290

87.59

(83.23-91.15)

6.77

P<0,0001

1.10

No

1264

3.96

(3.01-5.18)

1

No

657

12.94

(10.59-15.72)

1

3.27

Unknown

1392

8.91

(7.52-10.52)

2.25

Unknown

244

26.64

(21.20-32.65)

2.06

2.99

Reviewer 2 Report

In this paper, de Visscher et al. presented their serological diagnostic study on SARS-CoV-2 seroprevalence among health care workers in a non-academic Belgian hospital managing COVID-19 inpatients. The study design is that of a cross-sectional study and 4,008 tests were performed on 2,817 subjects. Comparisons were made on the two different sampling time points, i.e. Summer 2020 and Winter 2021. Different test kits under DiaSorin were used for the two phases, although the authors justified this by stating concordane studies had been done and demonstrated no difference in interpretation.

This paper is generally well-written and decently presented. Yet, some of the arguments in this paper are contentious. They would be separately discussed in below.

Major comments -

1. Please describe how recruitment bias could have affected the results. How was sampling performed? (this could potentially also affect interpretation of the results e.g. reversal of seroprevalence ratios) Also noteworthy would be the necessary recall bias - esp. with regard to the risk factors asked. The ORs obtained from self-reported risk factors could have been greatly inflated. 

2. Please detail the context of the study (i.e. vaccination campaign in Belgium- when started and on which groups first, whether mandatory in HCW etc.) to help readers understand the observations. How was the community transmission like during the study period?

3. Seroprevalence in Winter 2021 was higher than that of Summer 2020, as may be expected. According to the vendor (liaisonr_sars-cov-2_trimerics_igg_assay_m0870004408_a_lr_0.pdf (diasorin.com)), "Trimeric Spike Glycoprotein is a stabilized trimer offering an improved detection of IgG Neutralizing antibodies". Please provide evidence and data to either demonstrate equivalence of the new test to the older S1/S2 IgG test, or perform appropriate experiment to estimate how much of the increased seroprevalence was due to the purportedly "improved detection". 

Minor comments -

1. Ln 60-61: please provide clearer justification for "even when the PCR tests are available, they have a limited sensitivity", e.g. the stage of infection, the specific sample type

2. Throughout the manuscript: European decimal place "," to be aligned with MDPI / International convention

3. The manuscript would benefit from having salient seroprevalence comparisons highlighted in relevant bar charts. 

Author Response

B1. Please describe how recruitment bias could have affected the results. How was sampling performed? (this could potentially also affect interpretation of the results e.g. reversal of seroprevalence ratios) Also noteworthy would be the necessary recall bias - esp. with regard to the risk factors asked. The ORs obtained from self-reported risk factors could have been greatly inflated.

Thank you for this comment. We have added an additional explanatory text inserted from line 236 to line 247 in the discussion part of the manuscript.

In addition, please find our detailed explanation below for the reviewer :  

  1. Sampling :

  1. All collaborators of the Hospital have been proposed to undergo a serological test on SARS-CoV-2 IgG antibodies
  2. In Summer 2020, out of a total of 4500 collaborators, 3474 expressed their interest, but only 2817 came, on a voluntary basis, to be serologically tested.
    This sample can be considered as a “convenience sample” corresponding to more than 77 % and 63 % of the population respectively
  3. In Winter 2020-2021, the second phase of the study was operationally coupled with a vaccination campaign which was also proposed to the collaborators without obligation, and
    the same 4500 collaborators were proposed a second serological testing, without any obligation to accept one or the other proposal.
    2084 collaborators came on a voluntary basis, but, due to calendar constraints, 893 had already been injected a first dose of Pfizer Comirnaty vaccine 3 weeks before the blood sampling, and they are not included in this paper’s presentation.
    The other 1191 collaborators concerned by this paper were sampled the same day as the first vaccine injection, or they did not accept the vaccination proposal.
    This sample includes both participants  and non-participants to the first phase of the study, leading to the constitution of two independent samples.

  1. Biases :
    1. Sampling biases :

      1. First phase :

        The different parameters (risk factors) have been compared in the sample (N= 3474 with the intention to participate) and in the target population. Age and gender were similarly distributed, as mentioned in the results section.
        Other indicators (hospital profession, contact with patients, working on a specific site of the hospital, …) have also been compared without showing difference from the population structure.
        Additionally, Table 1 presents the comparison between this sample ant the subsample of health workers having undergone the blood sampling (N = 2817) : it clearly refutes the possibility of a non negligible selection bias at this level.
        In conclusion, we may really consider that our sample is fairly representative of the target population.

  1. Second phase :

The second phase only concerns the participants who have not been injected a first dose of vaccine before the blood sampling.

Table 1 shows no important differences between the two samples for what concerns age and gender.
But, according to the will to give access to the vaccination in priority to the caregivers, and to the slight delay between the blood sampling and the vaccination calendar, a slight deficit in caregivers appears in the Winter sample (mostly nurses and nurse-aides) as compared to the Summer sample (see Table 1). This difference is highly significant for both comparisons (P<0.0001)

  1. Recall biases :

Different characteristics of the hospital collaborators have been included in the analysis, and their assessment relies upon the responder’s declarations. This concerns:

  • Age
  • Gender
  • Contact with patients
  • Profession
  • Close contact at work/in the surrounding
  • History of symptoms suggestive of Covid-19

When possible (age, gender, profession and contact with patients) the frequency distributions of these characteristics have been compared with their frequency in the population: in this case, no discrepancy has been observed allowing to infer a lack of reliability of the information provided by the respondents.

For the other ones (close contact at work/in the surrounding, history of symptoms…), it has to be noticed that the information has been provided before the blood collection, and even longer before the results of the serological tests were known by the respondent.
The presence of IgG antibodies in the respondents mentioning no symptoms of Covid-19 (respectively 3.96 and 12.94 %) are an argument in favor of an underestimation of such antecedents.
If we may accept the hypothesis that these informations do not necessarily represent the true situation of the respondents, in contrast, it is hard to imagine that the answers could have been influenced by the serological status itself, therefore discarding their potential confounding effect when analyzing their influence on the serological status.

  1. Conclusion

    1. The first phase sample can be considered to be fairly representative of the target population

    2. The second phase sample does not differ from the first one according to age and gender ; in contrast, it is significantly different according to the proportions of different professions, with a deficit in caregivers and collaborators having contacts with patients (= selection bias for descriptive comparisons of both samples)

  • For what concerns risk factors, a lack of reliability can be suspected in the answers concerning exposure to contacts and history of symptoms, probably leading to an underestimation both of the exposure to contacts and of the presence of symptoms suggestive of Covid-19. (= recall bias possible)

  1. The presence of the observed selection bias :
    1. has an impact on the global comparison of both samples according to the serological status (namely 10.72 % and 33.92 % respectively); taking into account that the deficit categories in the second phase sample are also the ones with the highest seropositivity prevalence, we may conclude that the observed difference would have been even greater in the absence of this bias
    2. is not suspected to have any impact on the other comparisons presented in Table 2 showing significant differences for most of them
    3. is possibly responsible of the reversal of the (non-significant) relationship observed between positivity and gender: the selection bias concerns indeed mostly nurses and nurse aides, who are in majority females, and also presenting higher positivity rates due to their profession

  2. The presence of a recall bias with a confounding effect on the analytical results presented can be ruled out according to the chronology of the collection of the information

B2. Please detail the context of the study (i.e. vaccination campaign in Belgium- when started and on which groups first, whether mandatory in HCW etc.) to help readers understand the observations. How was the community transmission like during the study period?

We have incorporated this additional text from line 102 to line 109 in the chapter study design

“The two vaccination campaigns were organised by the Belgian Health Authorities. They decided to first vaccinate the Healthcare Workers in hospitals and then staff in retirement homes.

Belgian early vaccination started in January 2021, GHdC in February 2021, following guidelines, AND access to vaccines and deliveries.

At GHdC, the vaccination was administrated first to healthcare workers directly in contact with COVID-19 positive patients. This was gradually extended to the other members of the institution.

Vaccine administration and the participation in serological testing were voluntary and independent for each GHdC healthcare worker.”  

B3. Seroprevalence in Winter 2021 was higher than that of Summer 2020, as may be expected. According to the vendor (liaisonr_sars-cov-2_trimerics_igg_assay_m0870004408_a_lr_0.pdf (diasorin.com)), "Trimeric Spike Glycoprotein is a stabilized trimer offering an improved detection of IgG Neutralizing antibodies". Please provide evidence and data to either demonstrate equivalence of the new test to the older S1/S2 IgG test, or perform appropriate experiment to estimate how much of the increased seroprevalence was due to the purportedly "improved detection".

Please see the excel file attached  “Answer for the evaluators - Comparison of Diasorin kits”

B4-1. Ln 60-61: please provide clearer justification for "even when the PCR tests are available, they have a limited sensitivity", e.g. the stage of infection, the specific sample type

We have incorporated this additional text from line 61 to line 69 in the chapter introduction

B5-2. Throughout the manuscript: European decimal place "," to be aligned with MDPI / International convention

Done in the text and in table 3.

B6-3. The manuscript would benefit from having salient seroprevalence comparisons highlighted in relevant bar charts.

It seems to us that the changes achieved through reviewing are quite clear. If, however, this type of chart is desired, we would be happy to receive a detailed request of what is required.

Round 2

Reviewer 2 Report

I would like to thank the authors for addressing my major concerns. I am satisfied with the explanations and additional detail included in this new version.

As for the bar chart presentation, it’s indeed optional. If the authors think their current presentation is effective enough, I am also fine with it.